# Sesquiterpenoids from the Florets of *Carthamus tinctorius* (Safflower) and Their Anti-Atherosclerotic Activity

**DOI:** 10.3390/nu14245348

**Published:** 2022-12-16

**Authors:** Lei Li, Juan Liu, Xinrui Li, Yuqin Guo, Yunqiu Fan, Hongzhen Shu, Guangxu Wu, Cheng Peng, Liang Xiong

**Affiliations:** 1State Key Laboratory of Southwestern Chinese Medicine Resources, Chengdu University of Traditional Chinese Medicine, Chengdu 611137, China; 2School of Pharmacy, Chengdu University of Traditional Chinese Medicine, Chengdu 611137, China; 3Institute of Innovative Medicine Ingredients of Southwest Specialty Medicinal Materials, Chengdu University of Traditional Chinese Medicine, Chengdu 611137, China

**Keywords:** *Carthamus tinctorius*, sesquiterpenoids, structure elucidation, RAW264.7 cells, anti-atherogenic activity

## Abstract

(1) Background: The florets of *Carthamus tinctorius* L. are traditionally used as a blood-activating drug and can be used for the treatment of atherosclerosis, but no compounds with anti-atherosclerotic activity have been reported. (2) Methods: This study investigated the chemical compounds from the florets of *C. tinctorius*. Comprehensive spectroscopic techniques revealed their structures, and ECD calculations established their absolute configurations. Nile Red staining, Oil Red O staining, and cholesterol assessment were performed on these compounds and their aglycones for the inhibitory activity against the formation of foam cells induced by oxidized low-density lipoprotein (ox-LDL) in RAW264.7 macrophages. In addition, RAW264.7 macrophages were tested for their anti-inflammatory activity by measuring the inhibition of NO production caused by LPS. (3) Results: Five new sesquiterpenoids (**1**–**5**) isolated from the florets of *C. tinctorius* were identified as (–)-(1*R*,4*S*,9*S*,11*R*)-caryophyll-8(13)-en-14-ol-5-one (**1**), (+)-(1*R*,4*R*,9*S*,11*R*)-caryophyll-8(13)-en-14-ol-5-one (**2**), (–)-(3*Z*,1*R*,5*S*,8*S*,9*S*,11*R*)-5,8-epoxycaryophyll-3-en-14-*O*-*β*-D-glucopyranoside (**3**), (+)-(1*S*,7*R*,10*S*)-guai-4-en-3-one-11-*O*-*β*-D-fucopyranoside (**4**), and (–)-(2*R*,5*R*,10*R*)-vetispir-6-en-8-one-11-*O*-*β*-D-fucopyranoside (**5**). All compounds except for compound **3** reduced the lipid content in ox-LDL-treated RAW264.7 cells. Compounds **3** and **4** and their aglycones were found to reduce the level of total cholesterol (TC) and free cholesterol (FC) in ox-LDL-treated RAW264.7 cells. However, no compounds showed anti-inflammatory activity. (4) Conclusion: Sesquiterpenoids from *C. tinctorius* help to decrease the content of lipids, TC and FC in RAW264.7 cells, but they cannot inhibit NO production, which implies that their anti-atherogenic effects do not involve the inhibition of inflammation.

## 1. Introduction

Cardiovascular diseases have recently topped the list of causes of diseases in general, posing an immediate danger to human health. Atherosclerosis (AS) is a major pathological basis for many cardiovascular and cerebrovascular diseases, and macrophages play a major role in the plaque formation process of AS. In the early stages of AS development, macrophages phagocytose oxidize more low-density lipoprotein (ox-LDL) than they can metabolize, causing their massive accumulation of lipids and turning into foam cells. Then, foam cells gather together under the vascular endothelium to form a raised plaque, which further narrows or blocks blood vessels, thereby promoting the development of AS [1,2]. Therefore, inhibition of macrophage foaminess can effectively treat AS.

Recently, it has been demonstrated that many traditional Chinese medicines (TCMs), especially the blood-activating and stasis-removing medicines, are effective in treating AS [3,4,5]. The florets of *Carthamus tinctorius* L. (safflower), a well-known TCM widely cultivated in Xinjiang and Sichuan, are described as an “essential medicine for promoting blood circulation” in *Ben Cao Gang Mu*. Modern pharmacological studies have shown that safflower is effective in treating cardiovascular diseases and inhibiting the development of AS [6,7,8,9]. Additionally, safflower is utilized as food coloring, functional food, and feedstuff to supple fatty acids, improve hair health, extend endurance, and reduce wrinkles [10,11,12,13,14]. Up to now, phytochemical studies on safflower have revealed more than 200 chemical compounds, mainly including terpenoids, flavonoids, alkaloids, organic acids, and polyacetylenes [15,16,17,18]. However, fewer than 10 sesquiterpenoids have been reported [18]. In this study, five new sesquiterpenoids (compounds **1**–**5**) were identified from safflower, including caryophyllane-type, guaiane-type, and vetispirane-type sesquiterpenoids (Figure 1). Interestingly, compounds **4** and **5** are rare sesquiterpenoid fucopyranosides. To our knowledge, only a small number of fucopyranosides have been discovered in nature, and most of them exist as flavone glycosides [19,20], triterpenoid saponins [21,22], and steroidal saponins [23]. Their inhibitory actions on ox-LDL-induced lipid accumulation were explored by Nile Red staining, Oil Red O staining, and cholesterol ELISA testing to evaluate their anti-atherosclerotic activity.

## 2. Results and Discussion

### 2.1. Structure Elucidation of Compounds ***1**–**5***

Compound **1** is a colorless oil, and its molecular formula was determined as C_15_H_24_O_2_ with four degrees of unsaturation according to HR-ESI-MS at *m*/*z* 259.1666 [M+Na]^+^ (calcd for C_15_H_24_O_2_Na, 259.1674). The ^1^H NMR data (Table 1) of compound **1** exhibit characteristic signals of a hydroxymethyl [*δ*_H_ 3.24 (2H, d, *J* = 11.4 Hz, H_2_-14)], two methyls [*δ*_H_ 0.97 (3H, s, H_3_-15), 1.03 (3H, d, *J* = 7.2 Hz, H_3_-12)], and a terminal double bond [*δ*_H_ 5.05 (1H, brs, H-13a), 5.01 (1H, brs, H-13b)]. The ^13^C NMR (Table 2) and DEPT data of compound **1** detected 15 carbon signals in total, including two methyls, seven methylenes (one oxygenated and one olefinic), three methines, and three quaternary carbons (one ketonic and one olefinic). Based on these data, compound **1** is a caryophyllane-type sesquiterpenoid similar to gibberosin P, but the hydroxymethyl functionality at C-11 in compound **1** takes the place of a 4-hydroxyvaleryl group in gibberosin P [24]. The planar structure of compound **1** was confirmed with the assistance of ^1^H–^1^H COSY and HMBC correlations as shown in Figure 2. In particular, the hydroxymethyl unit is located at C-11 based on the HMBC cross-peaks from H_2_-14 to C-1, C-10, C-11, and C-15. An NOESY experiment was applied to identify the relative configuration of compound **1**. Correlations of H-9/H_3_-15 and H-1/H-4 and H_2_-14 (Figure 3) suggested that H-9 and Me-11 were co-facial, whereas H-1, H-4, and hydroxymethyl-11 occupied the opposite face. By comparing the computed electronic circular dichroism (ECD) data of compound **1** with the experimental ECD data (Figure 4), the absolute configuration of compound **1** was revealed to be 1*R*,4*S*,9*S*,11*R*. Therefore, compound **1** was determined to be (–)-(1*R*,4*S*,9*S*,11*R*)-caryophyll-8(13)-en-14-ol-5-one.

Compound **3** has a molecular formula of C_21_H_34_O_7_, as determined by the HR-ESI-MS. The ^1^H, ^13^C, and DEPT NMR data of compound **3** suggested that it is also a caryophyllane-type sesquiterpenoid. The signals for an anomeric proton [*δ*_H_ 4.24 (d, *J* = 7.8 Hz)] and an anomeric carbon (*δ*_C_ 104.7) revealed the existence of a *β*-glucosyl [25], indicating that compound **3** is a caryophyllane glycoside. In addition, the ^1^H and ^13^C NMR data (Table 1 and Table 2) exhibited signals attributed to an olefinic methine [*δ*_H_ 5.38 (1H, m, H-3), *δ*_C_ 124.1], an oxygenated methine [*δ*_H_ 4.50 (1H, dd, *J* = 9.6, 5.4 Hz, H-5), *δ*_C_ 82.0], an oxygenated quaternary carbon (*δ*_C_ 87.3), and an olefinic quaternary carbon (*δ*_C_ 140.8) in the aglycone unit. Thus, the existence of an epoxy moiety and a trisubstituted double bond was deduced based on the molecular formula. Comparison of the NMR data between compound **3** and 5*α*,8*α*-epoxycaryophyll-3-ene [26] suggested that an additional *β*-glucose moiety and an oxygenated methene group in compound **3** replaced a methyl group (C-14) in 5*α*,8*α*-epoxycaryophyll-3-ene, and this deduction was confirmed by the HMBC correlation from H-1′ to C-14.

In the NOESY spectrum of compound **3**, cross-peaks of H_2_-14/H-1 and H-9/H_3_-15 demonstrated the same orientation of H_2_-14 and H-1, whereas H-9 and H_3_-15 had the opposite orientation. The orientation of the 5-*O*-8 bridge was established by an NOE signal of H-9/H_3_-13. In addition, an NOE interaction between H-3 and H_3_-12 disclosed a *Z*-geometry for Δ^3^. To verify the absolute configuration of compound **3**, it was subjected to enzymatic hydrolysis, which yielded an aglycone **3a** and a D-glucose. Then, ECD calculation was carried out to resolve the absolute configuration of **3a**. As depicted in Figure 4, the calculated ECD spectrum of (1*R*,5*S*,8*S*,9*S*,11*R*)-**3a** and the experimental ECD spectrum of **3a** were in good agreement. Therefore, it was concluded that compound **3** is (–)-(3*Z*,1*R*,5*S*,8*S*,9*S*,11*R*)-5,8-epoxycaryophyll-3-en-14-*O*-*β*-D-glucopyranoside.

The molecular formula of compound **4** was identified as C_21_H_34_O_6_ by HR-ESI-MS at *m*/*z* 405.2244 [M+Na]^+^ (calcd for 405.2253). The ^1^H NMR spectrum (Table 3) exhibited distinguishable signals of five oxygenated methines [*δ*_H_ 4.42 (1H, d, *J* = 7.8 Hz, H-1′), 3.58 (1H, q, *J* = 6.6 Hz, H-5′), 3.56 (1H, d, *J* = 3.6 Hz, H-4′), 3.46 (1H, dd, *J* = 9.6, 3.6 Hz, H-3′), and 3.42 (1H, dd, *J* = 9.6, 7.8 Hz, H-2′)], which might be attributable to a *β*-glycosyl moiety. Additionally, five methyl groups [*δ*_H_ 0.64 (3H, d, *J* = 7.2 Hz, H_3_-14), 1.09 (3H, d, *J* = 6.6 Hz, H_3_-6′), 1.21 (3H, s, H_3_-12), 1.33 (3H, s, H_3_-13), and 1.66 (3H, s, H_3_-15)] were deducible from the ^1^H NMR data. The ^13^C NMR (Table 2) and the DEPT data revealed 21 carbon resonances attributed to the above units, four aliphatic methylenes, three aliphatic methines, and four quaternary carbons [an *α*,*β*-unsaturated ketone unit (*δ*_C_ 211.3, 138.3, and 181.1) and an oxygenated quaternary carbon (*δ*_C_ 81.3)]. Comprehensive analysis of the ^1^H-^1^H COSY, HSQC, and HMBC spectra indicated that the planar structure of compound **4** is similar to (1*R*,7*R*,10*S*)-11-*O*-*β*-D-glucopyranosyl-4-guaien-3-one [27], except for the glycosyl unit C-11. The glycosyl unit in compound **4** was further determined to be *β*-D-fucosyl due to the coupling constants (*J*_1′,2′_ = 7.8 Hz, *J*_2′,3′_ = 9.6 Hz, *J*_3′,4′_ = 3.6 Hz, and *J*_4′,5′_ ≈ 0 Hz), together with the ^13^C NMR data (*δ*_C_ 98.9, 72.5, 75.3, 73.1, 71.4, and 16.8) [20,28]. The NOESY correlations of H-1/H-7, H-1/H-9b, H_3_-14/H-9a, and H_3_-14/H-2b (Figure 3) were used to ascertain the relative structure of the aglycone moiety in compound **4**. Enzymatic hydrolysis of compound **4** provided the aglycone **4a**. Comparison of the experimental and calculated ECD spectra of **4a** (Figure 4) assigned 1*S*,7*R*,10*S* configuration to **4a**. Consequently, compound **4** was determined to be (+)-(1*S*,7*R*,10*S*)-guai-4-en-3-one-11-*O*-*β*-D-fucopyranoside.

The HR-ESI-MS of compound **5** revealed that it is a structural isomer of compound **4**. Comparative analysis of their ^1^H and ^13^C NMR data exhibited that they share similar functional groups such as an *α*,*β*-unsaturated ketone unit, four methyl groups, and a *β*-D-fucosyl moiety. However, as shown in Figure 2, 2D NMR data analysis pointed out that compound **5** is a vetispirane-type sesquiterpenoid [29]. The NOESY correlations of H_3_-15/H-1a, H-10/H_2_-4, and H-2/H_3_-15 (Figure 3) determined the relative structure of compound **5**. Similar to compound **4**, the aglycone **5a** was produced by enzymatic hydrolysis of compound **5**, and 2*R*,5*R*,10*R* configuration was assigned to **5a** according to ECD calculations (Figure 4). Therefore, the structure of compound **5** was established as (−)-(2*R*,5*R*,10*R*)-vetispir-6-en-8-one-11-*O*-*β*-D-fucopyranoside.

### 2.2. Anti-Atherosclerotic Activity of Compounds ***1**–**5*** and Aglycones ***3a**–**5a***

The anti-atherosclerotic activity of the isolates (**1**–**5**) and their aglycones (**3a**–**5a**) was evaluated by detecting their inhibitory effects against RAW264.7 cell foaminess induced by ox-LDL (Yiyuan Biotechnologies, Guangzhou, China) [1,30]. First, a CCK-8 assay was applied to evaluate the effects of the eight compounds on RAW264.7 cell viability, and all of the compounds were found to be noncytotoxic. Then, they were tested for anti-macrophage foaming activity by reducing the content of lipid in RAW264.7 cells treated by ox-LDL. As depicted in Figure 5, all compounds, except for compound **3**, showed significant activity by Nile Red staining. In particular, the effects of compounds **1**, **2**, **3a**, **4a**, **5**, and **5a** were equal to or superior to the effect of the positive control (25 μM of simvastatin). Interestingly, all of the aglycones (**3a**–**5a**) had better effects than their glycosides (**3**–**5**), which suggests that glycosidation of the sesquiterpenoids of safflower may result in a decrease in their anti-atherosclerotic activity. Although compound **4** showed weak activity in Nile Red staining, a significant effect was observed for compound **4** in Oil Red O staining (Figure 6). Specifically, compared with the control group, the model group collected more Oil Red O-stained lipid. In the drug group, compound **4** significantly decreased Oil Red O-stained lipid accretion.

To further explore the anti-atherosclerotic activity of compounds **3**, **3a**, **4**, and **4a**, the total cholesterol (TC) and free cholesterol (FC) levels in foamy macrophages were assessed by ELISA (Shanghai Keshun Science and Technology Co., Ltd., Shanghai, China). As shown in Figure 7, compounds **3**, **3a**, **4**, and **4a** significantly decreased the levels of TC and FC in ox-LDL-treated RAW264.7 cells in a dose-dependent way (*p* < 0.01 vs. model group). In addition, all compounds were investigated for their ability to inhibit lipopolysaccharide-induced NO production in RAW264.7 cells. The results showed that none of the compounds had an anti-inflammatory effect. Thus, it can be inferred that the safflower sesquiterpenoids do not exert their anti-atherogenic effects through the anti-inflammatory pathway [31]. Unfortunately, no further investigations on the molecular mechanisms were carried out due to the limited sample quantities of the isolates.

## 3. Experimental Section

### 3.1. General Experimental Procedures

General experimental instruments and materials are shown in Appendix A.

### 3.2. Plant Material

The florets of *C. tinctorius* were gathered from Luole Village, Sichuan Province, China. The sample was authenticated by Dr. Ji-hai Gao (Chengdu University of TCM, Chengdu, China). A voucher specimen (No. CT-20180608) was placed at the State Key Laboratory of Southwestern Chinese Medicine Resources, Chengdu University of TCM.

### 3.3. Extraction and Isolation

The florets of *C. tinctorius* (50 kg) were decocted twice with 12 times the quantity of H_2_O for 1 h each time. The extract was concentrated under reduced pressure in a big rotary evaporator (10 L) at 55 °C to produce a residue (16 kg). The residue was separated on a D-101 macroporous adsorbent resin (Sinopharm Chemical Reagent Co., Ltd., Shanghai, China) column using a gradient of H_2_O, 20%, 50%, 70%, and 90% EtOH to afford five fractions (A–E). Fraction D (66.7 g) was performed on a silica gel column (Yantai Institute of Chemical Technology, Yantai, China) and eluted with a gradient of CH_2_Cl_2_/MeOH (1:0–0:1) to afford 14 major fractions (D_1_–D_14_). D_7_ (6 g) was chromatographed on RP-MPLC eluted with MeOH/H_2_O (20:80–100:0) to obtain 10 subfractions (D_7-1_–D_7-10_). Six sub-subfractions (D_7-3-1_–D_7-3-6_) were obtained from D_7-3_ by silica gel column chromatography with a gradient elution system of CH_2_Cl_2_/MeOH (50:1–1:1). D_7-3-1_ was further separated into D_7-3-1-1_–D_7-3-1-7_ by column chromatography over Toyopearl HW-40F (Tosoh Corp., Tokyo, Japan) using 50% MeOH as the eluent. Compound **4** (3.5 mg) was derived from D_7-3-1-2_ by RP semipreparative HPLC (75% MeOH in H_2_O). D_7-3-1-3_ was further separated by TLC (CH_2_Cl_2_/MeOH, 10:1), followed by RP semipreparative HPLC (75% MeOH in H_2_O), to obtain compound **5** (2 mg). Separation of D_7-3-1-4_ by semipreparative HPLC (MeOH/H_2_O, 75:25) yielded compounds **1** (0.8 mg) and **2** (0.7 mg). D_7-3-2_ was separated by silica gel column chromatography and eluted with a gradient elution of CH_2_Cl_2_/MeOH (200:1–1:1) to yield D_7-3-2-1_–D_7-3-2-6_. From D_7-3-2-5_, compound **3** (3.0 mg) was obtained by TLC using CH_2_Cl_2_–MeOH (10:1) as the developing solvent.

### 3.4. Spectroscopic Data

(−)-(1*R*,4*S*,9*S*,11*R*)-Caryophyll-8(13)-en-14-ol-5-one (**1**): colorless oil; ${\left[\alpha \right]}_{D}^{20}$ − 4.2 (*c* 0.02, MeOH); IR *ν*_max_ 3437, 2957, 2925, 2855, 1699, 1657, 1629, 1454, 1376, 1039 cm^−1^; ECD (MeCN) *λ*_max_ (Δ*ε*) 209 (+3.70), 282 (+0.47), 326 (+0.11) nm; ^1^H NMR (CD_3_OD, 600 MHz) and ^13^C NMR (CD_3_OD, 150 MHz) data, see Table 1 and Table 2; (+)-HR-ESI-MS *m*/*z* 259.1666 [M+Na]^+^ (calcd for C_15_H_24_O_2_Na, 259.1674). The original ^1^H NMR, ^13^C NMR, DEPT, HSQC, ^1^H-^1^H COSY, HMBC, NOESY, IR, and (+)-HR-ESI-MS spectra are shown in Appendix A.

(+)-(1*R*,4*R*,9*S*,11*R*)-Caryophyll-8(13)-en-14-ol-5-one (**2**): colorless oil; ${\left[\alpha \right]}_{D}^{20}$ + 7.14 (*c* 0.03, MeOH); IR *ν*_max_ 3444, 3420, 2957, 2928, 2855, 1699, 1629, 1451, 1378, 1083, 1034, 888 cm^−1^; ECD (MeCN) *λ*_max_ (Δ*ε*) 207 (+7.25), 295 (+1.62); ^1^H NMR (CD_3_OD, 600 MHz) and ^13^C NMR (CD_3_OD, 150 MHz) data, see Table 1 and Table 2; (+)-HR-ESI-MS *m*/*z* 259.1666 [M+Na]^+^ (calcd for C_15_H_24_O_2_Na, 259.1674). The original ^1^H NMR, ^13^C NMR, DEPT, HSQC, ^1^H-^1^H COSY, HMBC, NOESY, 1D NOE, IR, and (+)-HR-ESI-MS spectra are shown in Appendix A.

(−)-(3*Z*,1*R*,5*S*,8*S*,9*S*,11*R*)-5,8-Epoxycaryophyll-3-en-14-*O*-*β*-D-glucopyranoside (**3**): colorless oil; ${\left[\alpha \right]}_{D}^{20}$ − 15.0 (*c* 0.02, MeOH); IR *ν*_max_ 3400, 2964, 2923, 2865, 1637, 1454, 1417, 1381, 1078, 1039 cm^−1^; ^1^H NMR (CD_3_OD, 600 MHz) and ^13^C NMR (CD_3_OD, 150 MHz) data, see Table 1 and Table 2; (+)-HR-ESI-MS *m*/*z* 421.2196 [M+Na]^+^ (calcd for C_21_H_34_O_7_Na, 421.2202). The original ^1^H NMR, ^13^C NMR, DEPT, HSQC, ^1^H-^1^H COSY, HMBC, NOESY, 1D NOE, IR, and (+)-HR-ESI-MS spectra are shown in Appendix A.

(−)-(3*Z*,1*R*,5*S*,8*S*,9*S*,11*R*)-5,8-Epoxycaryophyll-3-en-14-ol (**3a**): colorless oil; ${\left[\alpha \right]}_{D}^{20}$ − 4.8 (*c* 0.04, MeOH); ECD (MeCN) *λ*_max_ (Δ*ε*) 211 (+0.29), 251 (+0.05) nm; ^1^H NMR (CD_3_OD, 600 MHz) data, see Table 1. (+)-HR-ESI-MS *m*/*z* 259.1667 [M+Na]^+^ (calcd for C_15_H_24_O_2_Na, 259.1674). The original ^1^H NMR and (+)-HR-ESI-MS spectra are shown in Appendix A.

(+)-(1*S*,7*R*,10*S*)-Guai-4-en-3-one-11-*O*-*β*-D-fucopyranoside (**4**): colorless oil; ${\left[\alpha \right]}_{D}^{20}$ +29.4 (*c* 0.02, MeOH); IR *ν*_max_ 3403, 2923, 2852, 1676, 1623, 1464, 1386, 1172, 1065 cm^−1^; ^1^H NMR (CD_3_OD, 600 MHz) and ^13^C NMR (CD_3_OD, 150 MHz) data, see Table 2 and Table 3; (+)-HR-ESI-MS at *m*/*z* 405.2244 [M+Na]^+^ (calcd for C_21_H_34_O_6_Na, 405.2253). The original ^1^H NMR, ^13^C NMR, DEPT, HSQC, ^1^H-^1^H COSY, HMBC, NOESY, IR, and (+)-HR-ESI-MS spectra are shown in Appendix A.

(+)-(1*S*,7*R*,10*S*)-Guai-4-en-11-ol-3-one (**4a**): colorless oil; ${\left[\alpha \right]}_{D}^{20}$ + 26.7 (*c* 0.03, MeOH); ECD (MeCN) *λ*_max_ (Δ*ε*) 234 (+1.54), 318 (−0.19), 324 (−0.20) nm; ^1^H NMR (CD_3_OD, 600 MHz) data, see Table 3. (+)-HR-ESI-MS at *m*/*z* 259.1673 [M+Na]^+^, (calcd for C_15_H_24_O_2_Na, 259.1668). The original ^1^H NMR and (+)-HR-ESI-MS spectra are shown in Appendix A.

(−)-(2*R*,5*R*,10*R*)-Vetispir-6-en-8-one-11-*O*-*β*-D-fucopyranoside (**5**): colorless oil; ${\left[\alpha \right]}_{D}^{20}$ − 28.2 (*c* 0.04, MeOH); IR *ν*_max_ 3303, 2972, 2925, 1665, 1467, 1381, 1170, 1070, 990 cm^−1^; ^1^H NMR (CD_3_OD, 600 MHz) and ^13^C NMR (CD_3_OD, 150 MHz) data, see Table 2 and Table 3; (+)-HR-ESI-MS at *m*/*z* 405.2250 [M+Na]^+^ (calcd for C_21_H_34_O_6_Na 405.2253). The original ^1^H NMR, ^13^C NMR, DEPT, HSQC, ^1^H-^1^H COSY, HMBC, NOESY, IR, and (+)-HR-ESI-MS spectra are shown in Appendix A.

(−)-(2*R*,5*R*,10*R*)-Vetispir-6-en-11-ol-8-one (**5a**): colorless oil; ${\left[\alpha \right]}_{D}^{20}$ − 15.0 (*c* 0.02, MeOH); ECD (MeCN) *λ*_max_ (Δ*ε*) 235 (−1.74) nm; ^1^H NMR (CD_3_OD, 600 MHz) data, see Table 3. (+)-HR-ESI-MS at *m*/*z* 259.1673 [M+Na]^+^ (calcd for C_15_H_24_O_2_Na, 259.1674). The original ^1^H NMR and (+)-HR-ESI-MS spectra are shown in Appendix A.

### 3.5. Enzymatic Hydrolysis of Compounds ***3**–**5***

Snailase (20 mg) was used to hydrolyze compound **3** in water (5.0 mL) for 48 h at 37 °C. The aglycone (compound **3a**, 0.8 mg) was extracted by EtOAc (4 × 5 mL) in a liquid–liquid extractor and purified by HPLC. Compounds **4** (2 mg) and **5** (1.2 mg) were enzymatically hydrolyzed as above to obtain compounds **4a** (0.8 mg) and **5a** (0.5 mg).

### 3.6. ECD Calculation

The details of ECD calculation of compounds **1**, **2**, and **3a**–**5a** are shown in Appendix A).

### 3.7. Cell Viability Assay

RAW264.7 cells were grown in DMEM supplemented with 10% FBS under a humidified environment at 37 °C and 5% CO_2_. The cells were planted in 96-well plates (3 × 10^4^ cells/well) and incubated with compounds **1**–**5** and **3a**–**5a** (25, 50, and 100 μM) for one day. Then, the cells were treated by CCK-8 solution (10 μL/well) and incubated for 1 h. Under 450 nm, the absorbance of each well was tested to evaluate the viability of the cells.

### 3.8. Nile Red Staining

RAW264.7 cells were placed in 96-well plates (3 × 10^4^ cells/well). Then, the cells were treated with ox-LDL (75 μg/mL), together with compounds **1**–**5** and **3a**–**5a** (25, 50, and 100 μM). After incubation for 24 h, the cells were kept for 30 min at 37 °C with 4% paraformaldehyde, then rinsed with PBS. The cells were hatched with freshly prepared Nile Red staining solution (2 μg/mL) for 30 min. Finally, PBS was used to wash the cells, and the fluorescence intensity was measured at 530 nm/590 nm.

### 3.9. Oil Red O Staining

RAW264.7 cells were placed in 24-well plates (3 × 10^5^ cells/well). Then, the cells were treated with ox-LDL (75 μg/mL) and compound **4** (25, 50, and 100 μM). After incubation for 24 h, the cells were kept for 30 min at 37 °C with 4% paraformaldehyde, followed by rinsing with PBS. The cells were washed with 50% isopropanol for 20 s, and incubated with fresh-filtered Oil Red O solution (60% saturated Oil Red O/40% deionized water) for 30 min. The cells were washed with alternate rinses of 50% isopropanol and PBS to remove the floating colors and photographed for observation by fluorescence microscopy. Finally, the cells were washed with isopropanol, and the absorbance was observed at 550 nm.

### 3.10. ELISA

RAW264.7 cells (1 × 10^6^/well) were placed into six-well culture plates and treated by the above method. The contents of TC and FC were measured using the ELISA kits.

## 4. Conclusions

In conclusion, five new sesquiterpenoids were isolated from safflower, including caryophyllane-type, guaiane-type, and vetispirane-type sesquiterpenoids. The sesquiterpenoid types characterized herein have not been reported from *C. tinctorius*, and compounds **4** and **5** are rare sesquiterpenoid fucopyranosides. The sesquiterpenoids showed significant anti-atherosclerosis activity by decreasing the contents of lipid, TC, and FC in ox-LDL-treated RAW264.7 cells. Altogether, these new sesquiterpenoids not only enrich the diversity of sesquiterpenoids in safflower but also explain the effective material basis of safflower in treating AS.

## Figures and Tables

**Figure 1 nutrients-14-05348-f001:**
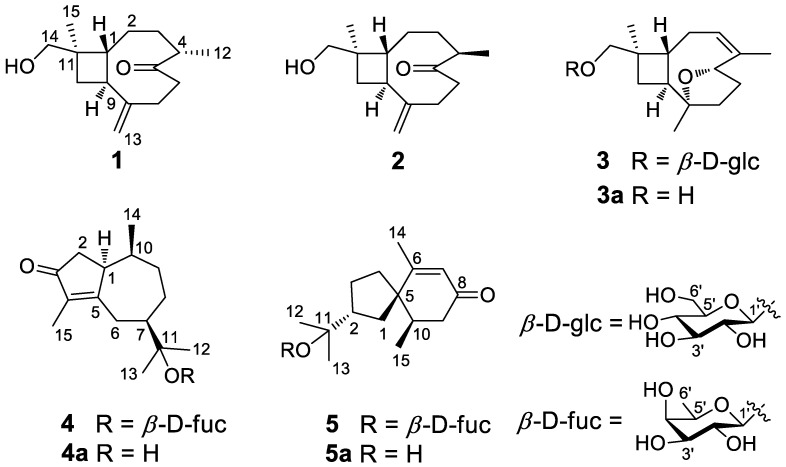
Structures of compounds **1**–**5** and **3a**–**5a**.

**Figure 2 nutrients-14-05348-f002:**
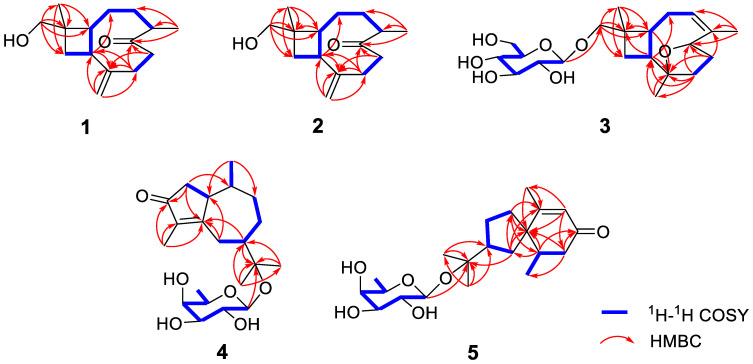
Key ^1^H–^1^H COSY and HMBC correlations of compounds **1**–**5**.

**Figure 3 nutrients-14-05348-f003:**
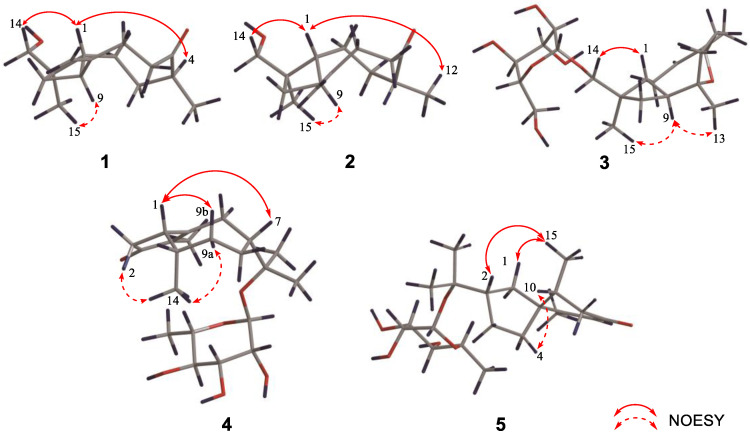
Key NOESY correlations of compounds **1**–**5**.

**Figure 4 nutrients-14-05348-f004:**
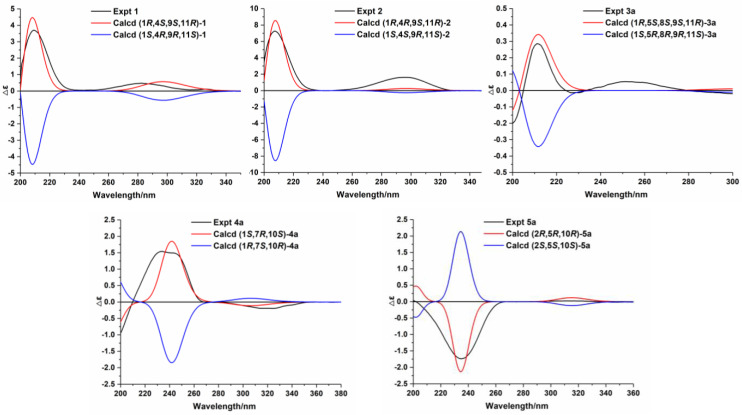
Experimental and calculated ECD spectra of compounds **1**–**2** and **3a**–**5a**. Based on the HR-ESI-MS data of compound **2**, its chemical formula is similar to that of compound **1**. The ^1^H, ^13^C, and 2D NMR spectra of compounds **2** and **1** showed that the planar structures of these two compounds are identical. Further analysis revealed that compound **2** was a C-4 epimer of compound **1** based on the NOESY correlations of H_2_-14/H-1, H-1/H_3_-12, and H-9/H_3_-15 (Figure 3). Moreover, ECD calculations showed that the absolute configuration of compound **2** is 1*R*,4*R*,9*S*,11*R* (Figure 4). Therefore, compound **2** was identified as (+)-(1*R*,4*R*,9*S*,11*R*)-caryophyll-8(13)-en-14-ol-5-one.

**Figure 5 nutrients-14-05348-f005:**
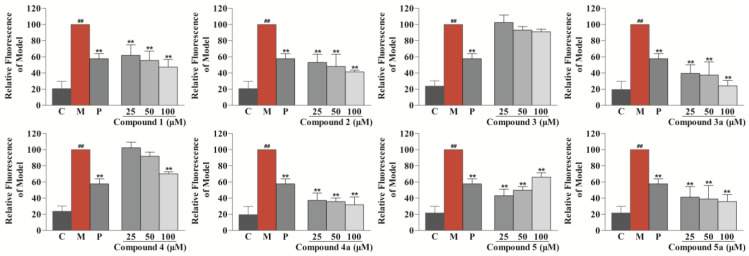
Effects of compounds **1**–**5** and **3a**–**5a** to inhibit lipid drops accumulation in ox-LDL-treated RAW264.7 cells determined by Nile Red staining. Results are expressed as the mean ± SD (*n* = 3). Model group was subject to 75 μg/mL ox-LDL. Positive group was subject to 25 μM simvastatin. ^##^ *p* < 0.01 vs. control group; ** *p* < 0.01 vs. model group. C: control group, M: model group, and P: positive group.

**Figure 6 nutrients-14-05348-f006:**
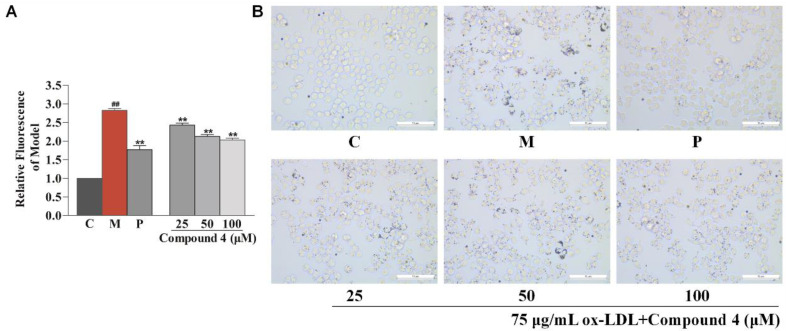
Inhibitory effects of compound **4** against lipid drops accumulation in ox-LDL-treated RAW264.7 cells measured by Oil Red O staining. (**A**) Oil Red O content evaluated by isopropanol extraction. (**B**) Oil Red O staining images of RAW264.7 cells (magnification 400×). Results are expressed as the mean ± SD (*n* = 3). Model group was subject to 75 μg/mL ox-LDL. Positive group was subject to 25 μM simvastatin. ^##^ *p* < 0.01 vs. control group; ** *p* < 0.01 vs. model group.

**Figure 7 nutrients-14-05348-f007:**
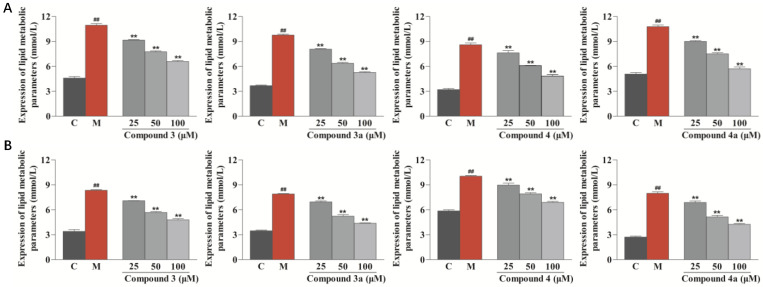
Effects of compounds **3**, **3a**, **4**, and **4a** on the contents of TC and FC in ox-LDL-treated RAW264.7 cells evaluated by ELISA. (**A**) Effects of compounds **3**, **3a**, **4**, and **4a** on the intracellular content of TC. (**B**) Effects of compounds **3**, **3a**, **4**, and **4a** on the intracellular content of FC. Results are expressed as the mean ± SD (*n* = 3). Model group was subject to 75 μg/mL ox-LDL. ^##^ *p* < 0.01 vs. control group; ** *p* < 0.01 vs. model group.

**Table 1 nutrients-14-05348-t001:** ^1^H NMR data of compounds **1**–**3** and **3a** in CD_3_OD (*δ* in ppm, *J* in Hz).

No.	1	2	3	3a
1	1.83 m	1.64 m	2.17 m	2.07–2.01 m
2	1.43 m 1.38 m	1.65 m 1.32 m	2.07 m	2.07–2.01 m
3	1.75 m 1.55 m	1.81 m 1.64 m	5.38 m	5.38 m
4	2.67 m	2.60 m		
5			4.50 dd (9.6, 5.4)	4.51 dd (9.6, 5.4)
6	2.64 m 2.39 m	2.62 m 2.50 m	2.49 m 2.01 m	2.49 m 2.07–2.01 m
7	2.48 m 2.37 m	2.48 m	2.37 m 1.61 m	2.30 m 1.61 m
9	2.23 m	2.46 m	2.11 m	2.07–2.01 m
10	1.79 dd (10.8, 10.2) 1.54 dd (10.8, 9.0)	1.73 dd (10.8, 10.2) 1.47 dd (10.2, 8.4)	1.69 dd (10.8, 10.2) 1.45 dd (10.2, 7.8)	1.52 dd (10.2, 10.2) 1.42 dd (10.2, 7.8)
12	1.03 d (7.2)	1.01 d (6.6)	1.66 s	1.65 s
13	5.05 brs 5.01 brs	4.93 brs 4.90 brs	1.18 s	1.18 s
14	3.24 d (11.4)	3.20 d (11.4)	3.68 d (10.2) 3.24 d (10.2)	3.25 d (10.8)
15	0.97 s	0.99 s	1.07 s	1.03 s
1′			4.24 d (7.8)	
2′			3.19 dd (9.0, 7.8)	
3′			3.33 t (9.0)	
4′			3.27 t (9.0)	
5′			3.27 ddd (9.0, 6.0, 2.4)	
6′			3.86 dd (12.0, 2.4) 3.67 dd (12.0, 6.0)	

**Table 2 nutrients-14-05348-t002:** ^13^C NMR data of compounds **1**–**5** in CD_3_OD (*δ* in ppm).

No.	1	2	3	4	5
1	49.6	47.7	41.0	47.2	32.5
2	27.2	28.8	30.9	42.3	52.2
3	31.2	32.3	124.1	211.3	28.8
4	48.6	49.0	140.8	138.3	35.9
5	219.4	220.0	82.0	181.1	51.7
6	42.9	43.4	35.5	35.3	173.3
7	35.7	33.0	32.2	47.3	125.9
8	155.1	154.0	87.3	28.5	202.4
9	44.1	44.5	50.4	38.0	43.6
10	35.6	34.3	31.7	36.8	38.5
11	39.3	39.8	39.0	81.3	79.2
12	17.2	17.2	26.3	22.8	26.2
13	112.4	112.1	26.7	25.2	24.7
14	71.0	71.2	78.1	12.5	21.3
15	17.6	17.6	17.0	7.9	16.8
1′			104.7	98.9	98.9
2′			75.8	72.5	72.6
3′			78.4	75.3	75.4
4′			71.7	73.1	73.1
5′			77.9	71.4	71.5
6′			62.8	16.8	17.1

**Table 3 nutrients-14-05348-t003:** ^1^H NMR data of compounds **4**, **4a**, **5**, and **5a** in CD_3_OD (*δ* in ppm, *J* in Hz).

No.	4	4a	5	5a
1	3.20 m	3.22 m	1.98 dd (13.8, 7.2) 1.67 dd (13.8, 12.0)	1.92 dd (13.8, 7.2) 1.55 dd (13.8, 12.0)
2	2.55 dd (18.6, 6.0) 2.03 d (18.6)	2.58 dd (18.6, 6.0) 2.03 d (18.6)	2.17 m	2.07 m
3			1.84 m	1.86 m
4			1.84 m	1.86 m
6	3.38 m 2.18 m	3.38 m 2.27 m		
7	1.95 m	2.04 m	5.74 brs	5.75 brs
8	1.92 m 1.30 m	1.95 m 1.30 m		
9	1.85 m 1.77 m	1.87 m 1.75 m	2.42 dd (16.8, 4.2) 2.24 dd (16.8, 9.6)	2.45 dd (16.8, 4.2) 2.24 dd (16.8, 9.6)
10	2.13 m	2.14 m	2.10 m	2.12 m
12	1.21 s	1.19 s	1.27 s	1.21 s
13	1.33 s	1.24 s	1.26 s	1.21 s
14	0.64 d (7.2)	0.64 d (7.2)	2.04 d (1.2)	2.02 d (0.6)
15	1.66 s	1.66 s	1.05 d (6.6)	1.04 d (6.6)
1′	4.42 d (7.8)		4.42 d (7.8)	
2′	3.42 dd (9.6, 7.8)		3.41 dd (9.6, 7.8)	
3′	3.46 dd (9.6, 3.6)		3.47 dd (9.6, 3.6)	
4′	3.56 d (3.6)		3.59 d (3.6)	
5′	3.58 q (6.6)		3.62 q (6.0)	
6′	1.09 d (6.6)		1.24 d (6.0)	

## Data Availability

The data presented in this study are available in the Appendix A or can be provided by the authors.

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
