# Peer review of "Sesquiterpenoids from the Florets of Carthamus tinctorius (Safflower) and Their Anti-Atherosclerotic Activity"

_nutrients, 2022, doi:10.3390/nu14245348_

Round 1
Reviewer 1 Report
This is an interesting paper. However, the following points should be addressed.
Major points:
1. The isolation and identification works of those compounds are straightforward. However, the authors should include the HPLC analysis results to measure their concentrations in the sample, Carthamus tinctorius, used in this study.
2. The anti-inflammatory activity of Carthamus tinctorius crude extracts should also be included.
3. Finally, evaluate each compound's role in anti-inflammation in vivo according to their content in Carthamus tinctorius.
Minor points:
I am unsure whether this paper's scope fits this journal's audience. Because Carthamus tinctorius (safflower) is generally used as medicine rather than a food or nutrition supplement, this paper is more suitable for publication in journals in medicinal chemistry, pharmacology, or alternative / complementary medicine.
Author Response
Major points:
Comment 1
The isolation and identification works of those compounds are straightforward. However, the authors should include the HPLC analysis results to measure their concentrations in the sample, Carthamus tinctorius, used in this study.
-> Thanks for the good comment. Our research mainly focuses on finding trace novel bioactive compounds of safflower. Less than 5 mg of sesquiterpenoids were isolated from safflower, indicating that the content of these trace sesquiterpenoids was very low. In addition, the chemical composition of safflower is complex, so we think that it is difficult to determine the content of these trace sesquiterpenoids by HPLC analysis. This study first discovered anti-atherosclerotic sesquiterpenoids from safflower, including two rare sesquiterpenoid fucopyranosides.
Comment 2
The anti-inflammatory activity of Carthamus tinctorius crude extracts should also be included.
-> Special thanks for this good suggestion. In our study, the crude extract was subjected to chromatography over a D-101 macroporous adsorbent resin column and eluted successively with H2O, 20% EtOH, 50% EtOH, 70% EtOH, and 90% EtOH to afford the corresponding fractions (A–E). The anti-inflammatory activity of all fractions was assayed, and the 70% EtOH elution portion (fraction D) showed the best activity (Figure R1). Thus, we explored the active compounds from fraction D in this study. Interestingly, the isolated compounds showed anti-atherogenic activity, but they cannot inhibit NO production, which implies that their anti-atherogenic effects do not involve the inhibition of inflammation. Because none of the compounds showed an anti-inflammatory effect, and this paper mainly focuses on the anti-atherogenic activity of safflower compounds, we have not added the anti-inflammatory activity of the crude extract in the manuscript.
Figure R1. Effects of different elution portions of safflower crude extract over a D-101 macroporous adsorbent resin column to inhibit LPS-induced NO production in RAW264.7 cells. Results are expressed as the mean ± SD (n=3). Model group was subject to 1 μg/mL LPS. ##P < 0.01 vs. control group; **P < 0.01 vs. model group.
Comment 3
Finally, evaluate each compound's role in anti-inflammation in vivo according to their content in Carthamus tinctorius
-> Thank you very much for the helpful suggestion. Unfortunately, the sesquiterpenoids from safflower were not examined for anti-inflammatory effects in vivo due to their limited quantities. We only assessed their anti-inflammatory and anti-atherogenic activities in vitro.
Minor points:
Comment 4
I am unsure whether this paper's scope fits this journal's audience. Because Carthamus tinctorius (safflower) is generally used as medicine rather than a food or nutrition supplement, this paper is more suitable for publication in journals in medicinal chemistry, pharmacology, or alternative / complementary medicine.
-> Thanks for your constructive suggestion. Safflower is also used as food coloring, functional food, and feedstuffs to supple fatty acids, improve hair health, extend endurance, reduce wrinkles and so on. We have added the description and references in the Introduction. In addition, this article is submitted to an interdisciplinary topic of the special issue "phytochemicals effect in chronic metabolic diseases".

Reviewer 2 Report
The manuscript refers to the isolation of sesquiterpenoids from Cartamus tictorium florets and contributes to the elucidation of the chemical composition of this medicinal/edible plant. The structural elucidation is well described, in my opinion. The Introduction can be improved, showing better the importance of the C. tinctorium florets as remedy and food.
Line 137-139:
Are the amount of plant material and crude extract correct (50 and 16 kg?)
Please, give more details about the extraction process. E.g., how much boiling water was used for the extraction? For water elimination, did you use lyophilization process or rotatory evaporation (in this case, please inform the used temperature)?
In the caption of Figures 5-7, please, describe what model group means.
Author Response
Comment 1
The Introduction can be improved, showing better the importance of the C. tinctorium florets as remedy and food.
-> Thanks very much for your helpful suggestion. Safflower is also used as food coloring, functional food, and feedstuffs to supple fatty acids, improve hair health, extend endurance, reduce wrinkles and so on. We have added the description and references in the Introduction.
Comment 2
Line 137-139: Are the amount of plant material and crude extract correct (50 and 16 kg?) Please, give more details about the extraction process. E.g., how much boiling water was used for the extraction? For water elimination, did you use lyophilization process or rotatory evaporation (in this case, please inform the used temperature)?
-> Special thanks for this good comment. The amounts of plant material and crude extract are correct. The result showed that the extraction rate of decoction of safflower was high. In addition, it is possible that a fraction of safflower was mushy during decoction, which was hard to be filtered. The details about the extraction process have been added to section 3.3 Extraction and Isolation.
Comment 3
In the caption of Figures 5-7, please, describe what model group means.
-> Thanks for the suggestion. We have added the description of the model group in the caption of Figures 5-7.